# Explaining attractive and repulsive biases in the subjective visual vertical

Stefan Glasauer [1]*, W. Pieter Medendorp [2]

**1** Computational Neuroscience, Brandenburg University of Technology Cottbus-Senftenberg, Cottbus, Germany, **2** Donders Institute for Brain, Cognition and Behaviour, Radboud University, Nijmegen, The Netherlands

* stefan.glasauer@b-tu.de

## Abstract

Perception of gravity can be assessed by measuring the subjective visual vertical (SVV), the visually indicated spatial direction that appears earth-vertical to an observer. When the SVV is assessed in darkness while the observer is roll-tilted, it shows substantial biases. At tilts larger than 45°, the bias is attractive, that is, the visual indicator appears vertical when rotated toward the observer. At smaller tilts, however, a repulsive bias is observed. The attractive bias has been explained within the Bayesian framework as the effect of a prior for upright posture. The repulsive bias has so far been considered anti-Bayesian, suboptimal, or as the result of uncompensated ocular counterroll. Here we show that both biases can be explained within a purely Bayesian model. More specifically, the repulsive bias at small roll-tilts is a consequence of the known tilt-dependent variability of the SVV, which is hypothesized to reflect different levels of sensory noise of the otolith organs. We thus provide a solution to a century-old question of why there is a repulsive bias in vertical perception.

## Author summary

To judge whether a picture frame on the wall is hanging vertically, we use the direction of gravity as a fundamental reference. Our estimate of this direction depends on multiple sensory cues, most prominently the signals from the vestibular organs in the inner ear. Yet, for more than a century, it has been known that our perception of vertical is not fixed but can be systematically biased by the tilt of the head. For large head tilts, our perceived vertical shifts towards our own head axis – an attractive bias known as the Aubert effect. In contrast, for small head tilts, our perceived vertical shifts away from the head axis – a repulsive effect referred to as the Müller effect. Over the past two decades, the attractive bias has been successfully explained within Bayesian frameworks as the optimal perceptual strategy to process uncertain and noisy sensory information. The repulsive bias has typically been considered as suboptimal or inconsistent with a Bayesian

provided the original author and source are credited.

**Data availability statement:** The code for the model has been published at G-Node (https://doi.org/10.12751/g-node.k52s4k) together with the digitized data from two figures in a published paper (Van Beuzekom & Van Gisbergen 2000), which is publicly available as PDF (doi: 10.1152/jn.2000.84.1.11) and which is cited accordingly.

**Funding:** This work was supported by the Deutsche Forschungsgemeinschaft (project number 532480418 to SG) and the Nederlandse Organisatie voor Wetenschappelijk Onderzoek (NWA-ORC-1292.19.298 to WPM, NWO-SGW-406.21.GO.009 to WPM), and Interreg North-West Europe (NWE-RE:HOME to WPM). The funders had no role in study design, data collection and analysis, decision to publish, or preparation of the manuscript.

**Competing interests:** The authors have declared that no competing interests exist.

interpretation. Here, by incorporating head-tilt-dependent sensory uncertainty, consistent with the known variation in sensory noise of the otolith organs, we show that both biases arise naturally within a single Bayesian framework.

## Introduction

Systematic biases in perception can be either attractive or repulsive with respect to the immediate spatial or temporal context (e.g. [1–3]). In rare cases, both types of biases exist in combination. Such a case occurs for the visual perception of the direction of gravity, the so-called subjective visual vertical (SVV). The SVV is experimentally assessed by the participant adjusting a visual line in front of her to the perceived direction of gravity while being tilted in various roll-tilt positions (Fig 1). From the very beginning of studying the SVV [5], researchers have noted that in the upright position, the mean SVV error is close to zero (Fig 1A), while with large tilt angles around 90°, the SVV tilts toward the participants own head axis and thus lies between the true gravitational vertical and the head-centered vertical (Fig 1C). In other words, it is attracted toward the head axis. However, for small tilt angles and some of the participants, the SVV is adjusted on the opposite side of the gravitational vertical, that is, it is repulsed from the head axis (Fig 1B). The attractive bias has been named Aubert or A-effect [5], the repulsive bias is referred to as Müller or [6], each after the researchers first reporting them. Even though the repulsive Müller effect has been found in many studies since (e.g. [7]) and was even demonstrated in non-human primates [8], the reason for this over-estimation of tilt is yet unclear. This is the focus of the present paper.

The attractive bias of the SVV belongs to the historically oldest descriptions of perceptual distortions. In contrast to many purely visual illusions (e.g., Zöllner or Poggendorf) described at about the same time, the SVV is also of interest in a clinical context for the diagnosis of lesions or damage to the vestibular system (e.g. [9]). Already early investigations of the SVV (e.g. [10–12]) concluded that the main sensory input to perception of orientation with respect to gravity (in the absence of visual cues) are the otolith organs, utricles and saccules, which are situated in the vestibular part of the inner ear, and measure linear acceleration of the head, including gravity.

One of the earlier attempts of modelling the SVV [13] was based on a vectorial hypothesis, where a head-fixed vector pointing in the direction of the head axis, called the idiotropic vector, was added to the gravity vector as measured by the otolith organs. The idiotropic vector alone yielded a good fit of the attractive A-effect but could not account for the E-effect. Therefore, Mittelstaedt [13] hypothesized that the relative weight of the two otolith organs, the utricles and the saccules, was not equal but larger for the utricles. Together with a normalization this hypothesis could account quite well for both the E-effect and the A-effect (Fig 2). Although this model was highly successful in quantitatively fitting the SVV, it does not offer a normative explanation of why perceptual biases of the SVV occur.

With the advent of Bayesian inference approaches and conceiving perception as a close-to-optimal estimation process, the view on apparent perceptual errors

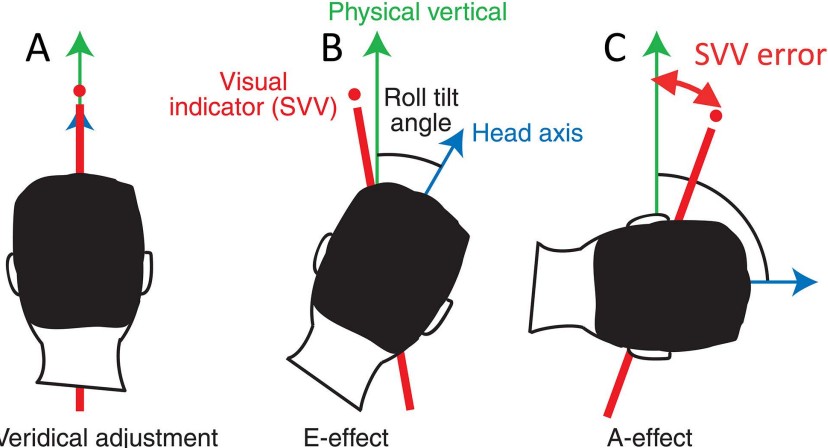

**Fig 1. Schematic representations of biases in the SVV task (modified from [4]). A**: no bias while upright. **B**: Small body tilt: repulsive bias (E-effect). **C**: Larger body tilt: attractive bias (A-effect). Error: angle between physical and subjective vertical direction.

and biases in the SVV also changed. The question was posed whether errors in the SVV are the result of a normative Bayesian estimation process, which aims to optimize the estimation in the presence of uncertainty such as sensory measurement noise. Within the Bayesian perspective, an obvious reason for an attractive bias such as the A-effect is that head-upright is the most common orientation with respect to gravity, which can be modelled as a prior distribution centered at roll-tilt zero, i.e., upright. Perceptual estimates would then be attracted towards this prior, resulting in the A-effect (see Fig 2). The first Bayesian model for the SVV was proposed by a student of Mittelstaedt [14], who successfully could explain the A-effect as consequence of a head-fixed prior for upright position. The Bayesian approach has since then been used by many authors to explain various features of the perception of gravity ([4,7,15–19], and many more). Despite this success, one problem has remained: explaining the repulsive bias of the E-effect within a Bayesian framework. The usual Bayesian approach predicts that estimated values are attracted towards the prior but not repelled by it. Eggert's model [14], for instance, attributed the E-effect to suboptimal integration of differences in measurement noise of the two

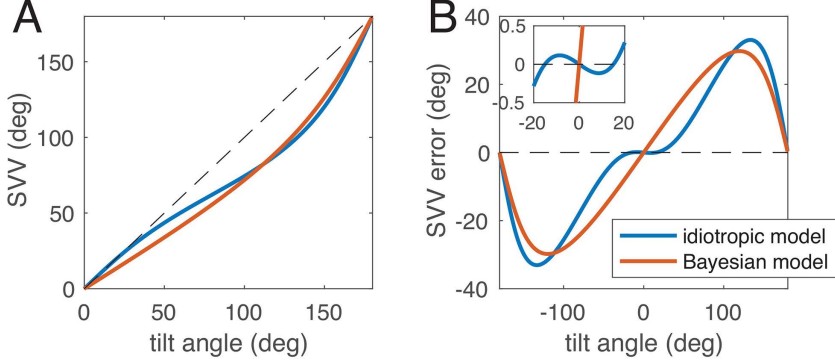

**Fig 2. The vectorial idiotropic model (blue) after Mittelstaedt [13] and a Bayesian model with a head-fixed prior for upright position.** The parameters of the Bayesian model (see Methods) have been adjusted to fit the idiotropic model. A: SVV plotted over tilt angle for roll tilt between upright (0°) and upside down (180°). The A-effect is clearly visible as underestimation of tilt for larger tilt angles. B: SVV error plotted over tilt angle for the full circle. The small repulsive E-effect simulated by the idiotropic model is visible in the inset.

otolith organs, without providing a complete Bayesian account. Other studies have linked the E-effect to an uncompensated effect of ocular counterroll (e.g. [16]) or neck proprioception [17] with the visual adjustment of the SVV, yet others have neglected it. Fig 2 shows a comparison between the original idiotropic model and a Bayesian model with a head-fixed prior to demonstrate that the A-effect, but not the E-effect, can be simulated. Consequently, repulsive biases such as the E-effect have also been called "anti-Bayesian" in the literature (e.g. [1,20,21]).

There have been different approaches to explain anti-Bayesian or repulsive biases in literature (e.g. [1,22,23]) using a variety of mechanisms, ranging from sensory adaptation over efficient coding to various flavors of probabilistic estimation, variably called Bayesian or predictive or optimal. However, none of these approaches have yet been applied and tested to explain the E-effect in the perception of the SVV.

In the following, we show that the E-effect can be explained as Bayes-optimal and in accordance with Bayesian perception if sensory uncertainty is tilt-dependent, which in turn leads to asymmetric likelihood functions. Tilt-dependent sensory uncertainty occurs if, as proposed previously [14], sensory noise for both otolith organs is assumed to differ. We also show that such a model is in accordance with the well-known anisotropy of the variability of the SVV (e.g. [7,17,24,25]).

## Methods

Participants in typical experiments are asked to adjust a visual indicator, usually a line, in an otherwise dark environment to the direction which they perceive as upright. This adjustment is performed in different roll-tilt body positions (see Fig 1). We therefore implemented a Bayesian model of the SVV, for which we considered as input the vestibular information from the otolith organs, which measures the direction of gravity, and as output the perceived tilt of the gravitational upright, which is provided as roll position of the visual indicator.

### The Bayesian model

For the Bayesian model we consider that the true roll-tilt angle of the head with respect to gravity (Fig 1) is unknown but can be measured by the vestibular sensors. However, this sensory measurement is corrupted by measurement noise. Therefore, the perceptual process aims to estimate the roll-tilt angle by combining prior knowledge about the distribution of head positions with knowledge of the measurement uncertainty. We assume that the prior knowledge for the angular head tilt can be represented as von-Mises distribution with mean 0 and dispersion $1/\kappa_h$. The measurement uncertainty comes from a von-Mises distribution with mean 0 and dispersion $1/\kappa_v$. Importantly, we assume that the measurement noise can be signal-dependent, that is, the dispersion $1/\kappa_v$ depends on the actual tilt angle.

The Bayesian estimation model for this case is formalized as follows. The prior distribution is given as von-Mises probability density function:

$$p(\theta) = \frac{1}{2\pi I_0(\kappa_h)} e^{\kappa_h \cos\theta}$$

(1)

with $\theta$ being the roll-tilt angle of the head and $I_0(\kappa_h)$ the modified Bessel function of the first kind of order 0 (required for normalizing the probability density).

The measurement distribution, i.e., the distribution of the measurement $\varphi$ for a given stimulus value $\theta$ is:

$$p(\varphi|\theta) = \frac{1}{2\pi I_0(\kappa_v)} e^{\kappa_v \cos(\varphi-\theta)}$$

The likelihood function is the measurement distribution viewed as function of the parameter [26], the unknown roll-tilt angle of the head $\theta$:

$$\mathcal{L}(\theta; \varphi) = \frac{1}{2\pi I_0(\kappa_v)} e^{\kappa_v \cos(\varphi - \theta)}$$

(2)

with $\varphi$ being the measured angle.

The posterior distribution can now be calculated by multiplying the prior distribution and the likelihood function, and normalizing it so that the integral over the posterior density becomes unity:

$$p\left(\theta | \varphi\right) \propto p(\theta) \cdot \mathcal{L}(\theta; \varphi)$$

(3)

Finally, as estimated or perceived tilt angle, the SVV, the angular expectation $\hat{\theta}$ of the posterior density $p\left(\theta | \varphi\right)$ is computed.

### Bayesian model 1: constant sensory noise

If the sensory noise is independent of roll tilt angle and thus $\kappa_v$ is constant, then the likelihood is a von-Mises probability density, and the resulting posterior distribution is again a von-Mises density. It is possible to analytically provide the posterior (see for example [27]), but we used numerical simulation as for all models. The two free parameters of this constant noise model are $\kappa_h$ and $\kappa_v$.

### Bayesian model 2: signal-dependent noise, full likelihood

For signal-dependent noise, the dispersion $1/\kappa_v$ depends on the true tilt angle $\theta$, so that the likelihood function becomes

$$\mathcal{L}(\theta; \varphi) = \frac{1}{2\pi I_0(\kappa_v(\theta))} e^{\kappa_v(\theta) \cos(\varphi - \theta)}$$

(4)

Note that in general this is no longer a von-Mises density. However, it becomes evident that an accurate likelihood function now requires complete knowledge of the noise characteristics at any possible tilt angle.

To implement signal-dependent noise, we consider the measurement of the tilt angle by the vestibular gravity sensors, utricles and saccules. The utricles measure the sine, the saccules the cosine of the roll tilt angle $\theta$ with respect to gravity. We assume that both sensors are affected by constant measurement noise $\varepsilon_x$ and $\varepsilon_z$ which is normally distributed with zero mean and variances $\sigma_x^2$ and $\sigma_z^2$. We allow the noise variances of utricles and saccules to be different as proposed previously [14]. The measured angle $\varphi$ is thus

$$\varphi = \tan^{-1} \frac{a_{utr}}{a_{sac}} = \tan^{-1} \frac{\varepsilon_x + \sin \theta}{\varepsilon_z + \cos \theta}$$

(5)

with measured accelerations $a_{utr}$ and $a_{sac}$ given in units of $g$ with $g = 9.81$ $m/s^2$.

Using a first-order Taylor series approximation for error propagation, the expectation of the measured angle $\varphi$ equals the actual roll angle $\theta$, and we can approximate the resulting variance of the angle $\varphi$, and therefore the noise dispersion $1/\kappa_v(\theta)$, as

$$1/\kappa_v(\theta) \approx \sigma_\varphi^2 \approx \sigma_x^2 \cos^2 \theta + \sigma_z^2 \sin^2 \theta$$

(6)

Using this approximation, the measurement noise is described by a von-Mises density with tilt-dependent dispersion $1/\kappa_v(\theta)$ and mean 0. Thus, if $\sigma_z^2 > \sigma_x^2$, the noise dispersion is minimal for 0° (upright) and 180° (upside down), with maxima

for 90° and -90° roll tilt. The signal-dependent noise model has three free parameters, the noise variances $\sigma_x^2$ and $\sigma_z^2$, and the concentration parameter of the prior $\kappa_h$.

### Bayesian model 3: signal-dependent noise, local likelihood approximation

Instead, we can also choose an approximative strategy by assuming that noise dispersion depends on the measured angle $\varphi$ rather than the true tilt angle $\theta$. In that case, only the local information about noise dispersion at the particular measured tilt angle is required. The resulting approximative likelihood function is again a von-Mises density, even though a different one for each measurement:

$$\mathcal{L}(\theta; \varphi) = \frac{1}{2\pi I_0(\kappa_v(\varphi))} e^{\kappa_v(\varphi)\cos(\varphi-\theta)}$$

(7)

with $1/\kappa_v(\varphi) \approx \sigma_x^2\cos^2\varphi + \sigma_z^2\sin^2\varphi$ and thus again three free model parameters.

### The idiotropic model

We used the idiotropic model of the SVV [13] as comparison, because it provides a good description of the known data and can thus be used as reference. It is given by the following equations. For consistency with Mittelstaedt's original publication, the SVV angle is denoted as $\beta$. The tilt angle $\theta$ is measured as otolith responses of the utricles $u$ and saccules $s$.

$$u = g \sin \theta$$

(8a)

$$s = f_z g \cos \theta$$

(8b)

$$N = \sqrt{u^2 + s^2}$$

(8c)

$$\beta = \tan^{-1} \frac{u/N}{s/N + M}$$

(8d)

where $g$ = 9.81 m/s$^2$ is gravitational acceleration, $f_z$ the saccular weighting, and $M$ the idiotropic vector. $N$ is a normalization factor. Mittelstaedt's original parametrization was $M$ = 0.4 and $f_z$ = 0.7, producing a large A-effect. For the simulation of the idiotropic model in Fig 2 and demonstration of the E-effect, we used these original parameters.

### Response variability

The variability of the posterior distribution is, in general, not equal to the variability of the distribution of behavioral responses [17,26]. While the variability of the response distribution can be analytically derived in the Gaussian case, this is not possible for the non-linear function relating tilt angle to SVV. However, there are two possible solutions: since the SVV dependence on tilt angle in the models is a deterministic function, i.e., $\hat{\theta} = f(\theta)$, the SVV response variance can be approximated via 1$^{st}$ order Taylor expansion by $\sigma_\theta^2 = \left(\frac{d}{d\theta}f(\theta)\right)^2 \cdot \sigma_\varphi^2$, with $\sigma_\varphi^2$ being the variance of the measurement noise (Eqn. 6). Alternatively, the variance of the SVV at a particular stimulus angle can be estimated by Monte-Carlo simulation. For fitting the response variability to the data, the Taylor expansion approximation is used, because Monte-Carlo estimation within a minimization is more time-consuming. However, we verified that for the final solution both methods yielded the same results. For the response variance shown in the Results, we chose the Monte-Carlo method and simulated the model n = 1e5 times for each tilt angle with noisy measurements according to Eqn. 5.

## Alternative prior densities

In addition to the von-Mises density described by Eqn.1 we used the following densities (see also [19]) as alternatives for the prior:

- Wrapped normal density with mean 0 and standard deviation $\sigma$

$$p(\theta) = \frac{1}{\sigma\sqrt{2\pi}} \sum_{k=-\infty}^{\infty} e^{\frac{-(\theta+2\pi k)^2}{2\sigma^2}}$$

- Mixture of uniform and normal density with mean 0 and standard deviation $\sigma$, with weight $w$ of the uniform part

$$p(\theta) = (1-w)\frac{1}{\sigma\sqrt{2\pi}} e^{\frac{-\theta^2}{2\sigma^2}} + \frac{w}{2\pi}$$

- Wrapped $t$ location scale density with mean 0, and two free parameters, the scale parameter $\sigma$ and parameter $\nu$

$$p(\theta) = \frac{\Gamma\left(\frac{\nu+1}{2}\right)}{\Gamma\left(\frac{\nu}{2}\right)\sigma\sqrt{\nu\pi}} \sum_{k=-\infty}^{\infty} \left(1 + \frac{(\theta+2\pi k)^2}{\nu\sigma^2}\right)^{-(\nu+1)/2}$$

with $\Gamma(\cdot)$ denoting the gamma function.

## Model simulation and fitting

Models were simulated numerically using Matlab (The Mathworks) according to the equations above with a discretization step of $2\pi/1000$ for representing densities and likelihoods. For fitting the Bayesian models to the idiotropic model and to the data, we used the Matlab function lsqnonlin. Differences between two angles $\varphi_1$ and $\varphi_2$, required for fitting, were calculated using complex numbers as $\arg\left(e^{i\varphi_1} \cdot e^{-i\varphi_2}\right)$. As mentioned above, the SVV angle was computed as angular expectation of the posterior density of the respective models. The standard deviations of the posterior and the response distributions were calculated as circular standard deviations.

For fitting the models to the experimental data, we simultaneously fitted the SVV adjustment for the full 360° together with the SVV standard deviation for tilt smaller than 135°, because with larger tilt angles, the SVV variability is known to be quite unreliable.

## Results

We first simulated the idiotropic model with original parameters $M = 0.4$ and $f_z = 0.7$ [13] as reference. We then fitted the SVV angles predicted by the three Bayesian models to the SVV angles of the idiotropic model. As shown in Fig 3A and 3B, the model with signal-dependent noise and full likelihood can simulate an E-effect and closely resembles the idiotropic model, unlike the other two Bayesian models. As expected from visual inspection, the root-mean-square error for the full likelihood model was much smaller (0.6°) than for the models with local likelihood approximation (5.3°) or with constant noise (6.0°).

Fig 3C shows the predicted standard deviation of the SVV for the three Bayesian models. The predictions have in common that the standard deviation should be small at the upright position and larger at tilted positions, which is indeed the case (e.g. [7,25]). The model parameters of the full likelihood model were $\frac{1}{\kappa_h} = 0.17$ (corresponding to an angular standard deviation of the prior 24.8°), utricular standard deviation $\sigma_x = 0.09$ g, and saccular SD $\sigma_z = 0.33$ g.

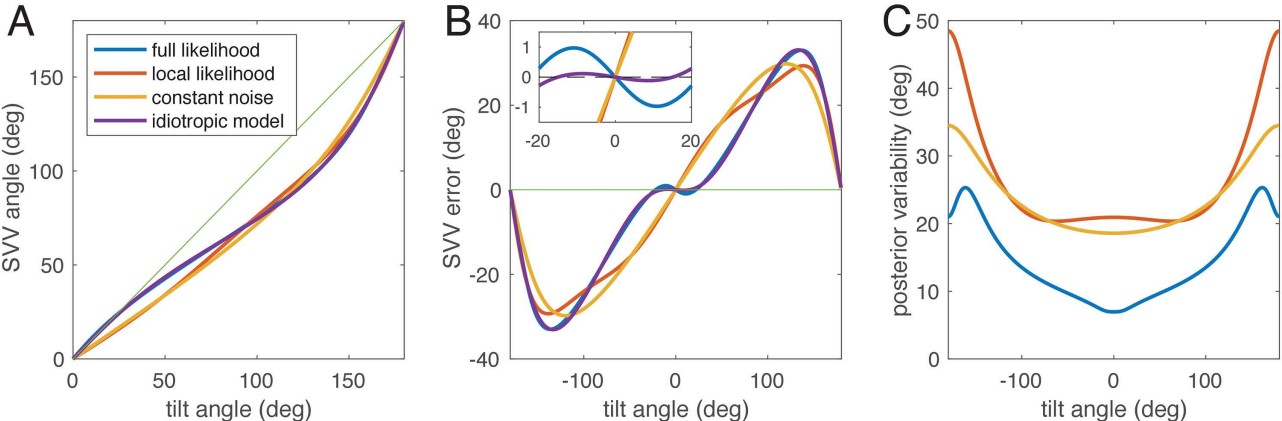

**Fig 3. Comparison of the idiotropic model (violet) after Mittelstaedt [13] and the best fits of the three Bayesian models to the idiotropic model.**
A: SVV angle plotted over tilt angle for roll tilt between upright (0°) and upside down (180°). B: SVV error plotted over tilt angle for the full circle. The
E-effect is visible for the idiotropic model and the Bayesian model with full likelihood, but not for the other two Bayesian models. Inset shows enlarged
region around tilt 0° to show the E-effect. C: predicted posterior variability (circular standard deviation) for all three Bayesian models. Without additional
assumptions, the idiotropic model does not offer a prediction for variability. Note that the actual variability of SVV responses can differ from the posterior
variability ([17,26], see also Fig 5).

To visualize the complete estimation process, inspired by Girshick et al. [23], Fig 4 depicts our full likelihood model,
using the parameters given above.

For a comparison with real data, we digitized Figs 4 and 11 of Van Beuzekom & Van Gisbergen [7], who assessed the
SVV and its variability in six subjects. We then fitted the full-likelihood model to the SVV data and the experimentally found
variability. Both results are shown in Fig 5. As can be seen from the figure, the fit captured the SVV data very well, and
the resulting response variability is not only in the same range as the SVV variability reported by Van Beuzekom & Van
Gisbergen [7], but also approximately matches the shape of the variability dependence on roll tilt. The model parameters
for this fit are the dispersion of the prior $\frac{1}{\kappa_h}$ = 0.097 (corresponding to a circular standard deviation of 18.3°), the utricular
standard deviation $\sigma_x$ = 0.027 $g$, and the saccular SD $\sigma_z$ = 0.216 $g$.

To understand what determines whether an E-effect occurs or not, the model comparisons in Fig 3 are helpful. The
E-effect could be fitted only with the full-likelihood model with signal-dependent noise, but not if the likelihood functions
were approximated by the local noise distributions. In Fig 6 we compare noise distributions with likelihood functions: while
noise distributions are symmetric, likelihood functions for small tilt angles are asymmetric (see also Fig 4).

For modelling the perceived SVV angle as result of a Bayesian estimation, the likelihood function is multiplied with the
prior distribution (Eqn. 3, and the mean of the resulting posterior distribution is considered the estimated SVV angle, see
also Fig 4). For both the local-likelihood model and the full-likelihood model, Fig 7 shows all three distributions for a tilt
angle of 10°. As evident from the figure, the resulting posterior distributions (yellow) are slightly different, and, more impor-
tantly, the mean of the posterior (dashed black line) is attracted towards zero for the local likelihood model (Fig 7A) and
thus smaller than the actual roll tilt, but it is repulsed from zero and larger than the tilt angle for the full-likelihood model
(Fig 7B).

The ability of the model to simulate the E-effect thus depends on the asymmetry of the likelihood functions, which
in turn depends on how much the measurement noise depends on the stimulus, i.e., the roll tilt angle. To visualize this
dependence, we changed parameter $\sigma_x^2$ of the model (see Eqn. 6), the variance of the utricular measurement noise, from
its fitted value to the same value as the saccular noise variance $\sigma_z^2$. Fig 8 shows simulations of the resulting models. As
evident from the SVV error, the model exhibits an E-effect only for strong stimulus dependence of the measurement noise.

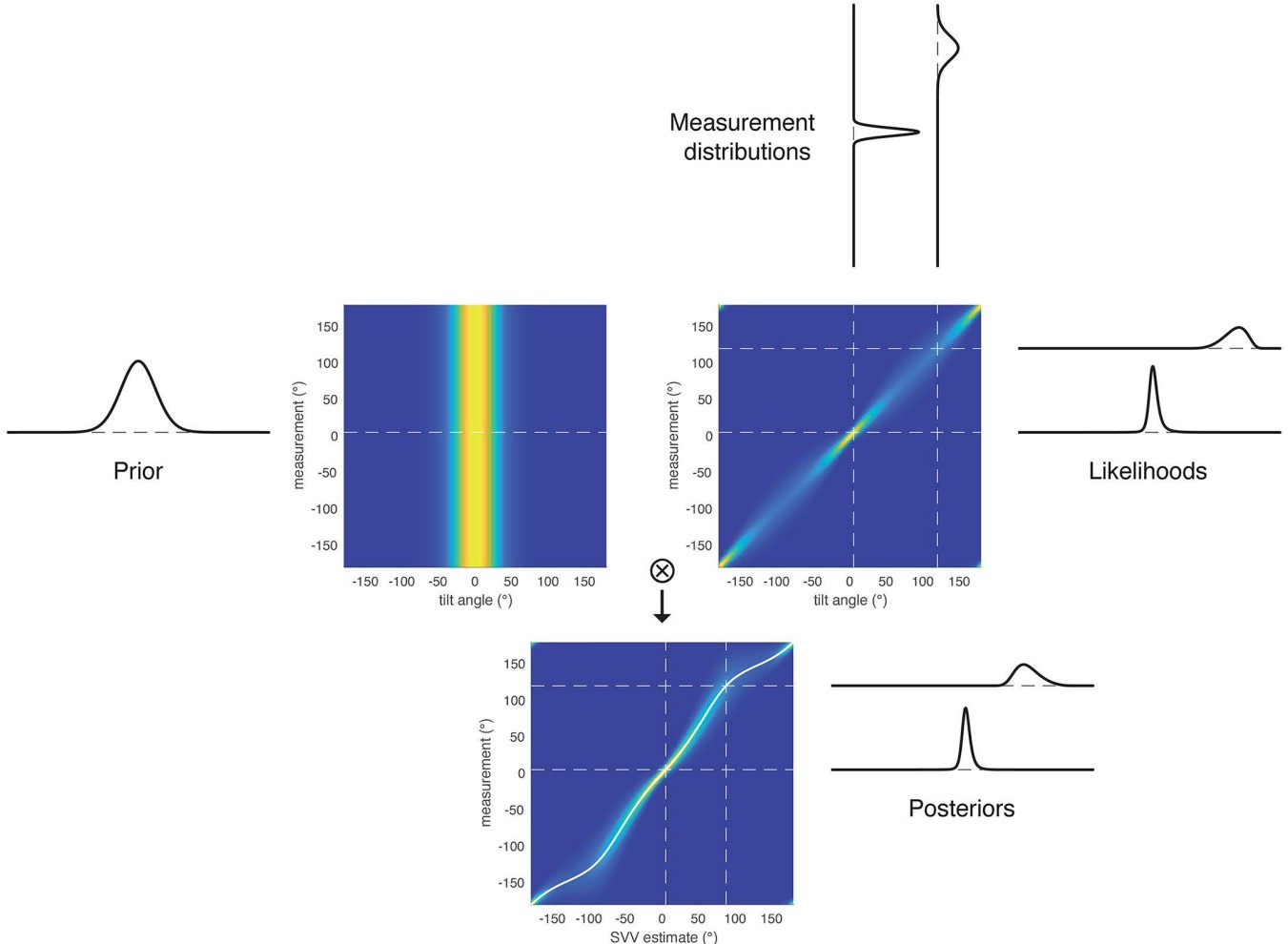

**Fig 4. Schematic for the Bayesian estimation process of the full likelihood model (visual presentation of the model inspired by [23]).** In the colored panels, the x-axis is physical tilt angle θ (upper panels) or estimated SVV angle θ̂ (lower panel) and the color corresponds to probability (white over yellow to blue: high to low). The upper left panel shows the prior $\mathbf{p}(\theta)$, which is independent of the measurement φ. The upper right panel shows the conditional probability $\mathbf{p}(\varphi|\theta)$. Vertical slices through the upper right panel show the symmetric measurement distributions, horizontal slices are the likelihood functions, which become asymmetric. Element-wise multiplications of these matrices (⊗) and row-wise normalization of the result yields the lower panel, which gives the posterior distributions $\mathbf{p}(\hat{\theta}|\varphi)$ as horizontal slices. The angular mean of the posterior distributions θ̂ is plotted as white line in the lower panel. Note that axes in this lower panel are flipped in comparison to Fig 3A. Slices are shown for tilt angles of 5° and 120°, parameters are computed from fitting the idiotropic model.

From the figure it can also be seen that the model can simulate SVV error with or without E-effect, depending on the relation between $\sigma_z^2$ and $\sigma_x^2$.

Thus, reducing the uncertainty in measuring acceleration along the head's z-axis might also reduce the repulsive bias or E-effect.

A further question is whether the experimental method used could influence the size of the observed E-effect. In the literature, two main methods are used to determine the SVV: the adjustment of a visual indicator to the perceived vertical, and the psychophysical 2AFC (two alternative forced choice) method. While, as argued above, a reasonable cost function for adjustment is quadratic, this is not true for 2AFC, for which the cost function can be modeled by a sign function,

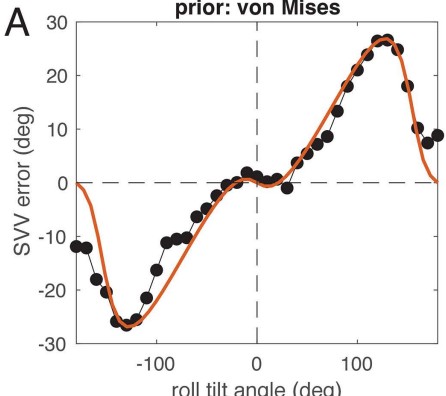
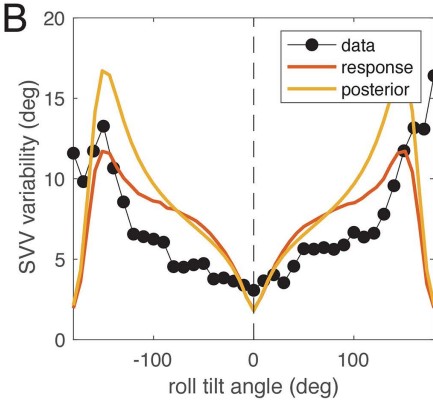

**Fig 5. The full-likelihood model (red) and data (black dots) from Van Beuzekom & Van Gisbergen [7].** A: SVV error (black) plotted over roll tilt angle together with best fit of the model (red). B: experimental SVV variability (black) plotted over roll tilt angle together with the fitted response variability (red) and variability of the posterior distribution (yellow) from numerical model simulation, both expressed as angular standard deviation.

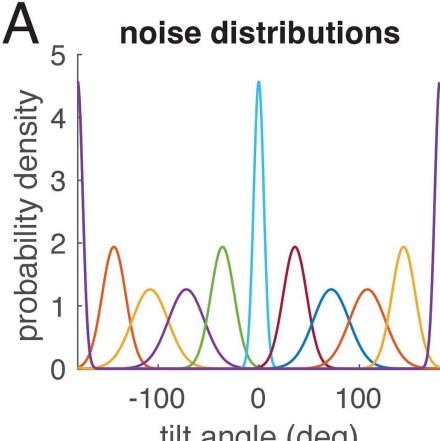
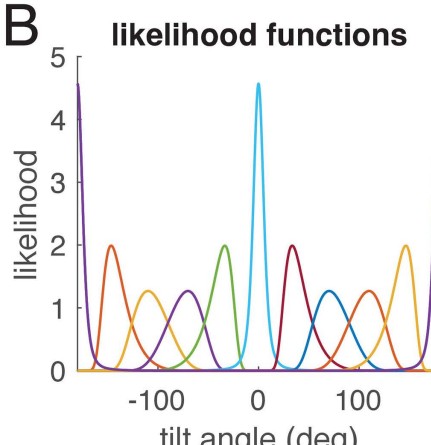

**Fig 6. Noise distributions(A) and corresponding likelihood functions (B) for the full-likelihood model shown in in Figs 3 and 4.** Each color shows one noise distribution centered at the tilt angle in A and the corresponding likelihood function in B. While the noise distributions are symmetric von-Mises distributions, the likelihood functions for small tilt angles are clearly asymmetric.

because the correct answer only depends on the sign of the angular difference between indicator and perceived vertical, but not on its magnitude. The theoretical value corresponding to such a cost function is the median of the distribution, because it is the value for which 50% of the distribution fall below. We also simulated the 2AFC numerically with Monte-Carlo simulation of 10000 adjustments per indicator line orientation using noisy stimuli and the parameters of the model fit in Fig 5. The results for head tilts of 5° and 90° are shown in Fig 9. Although the difference between mean and median is small in both cases, the 2AFC method leads to slightly smaller E-effect at 5° with the point of subjective equality (posterior median) lies closer to the stimulus than the posterior mean does. In contrast, for the 90° stimulus, the 2AFC method yields a slightly larger A-effect.

Finally, we examined the role of the prior distribution in model fitting following a previous investigation [19], in which it was found that the natural statistics of head tilt are closer to a *t* location-scale distribution than to a normal distribution. We

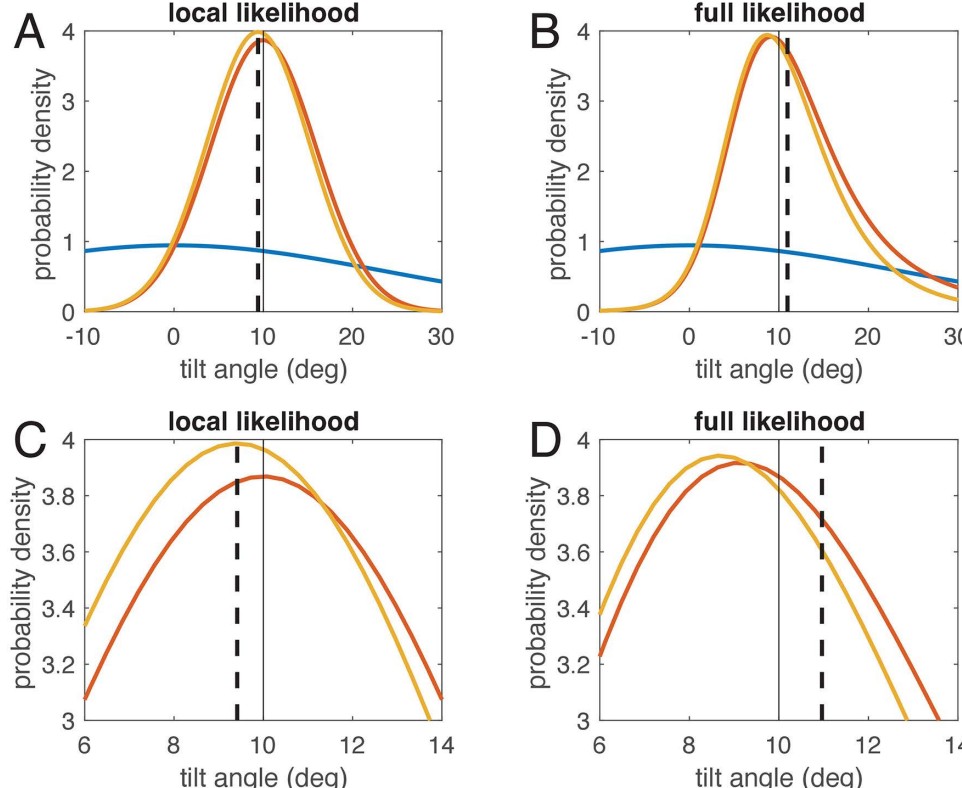

**Fig 7. Bayesian estimation for two models: to estimate tilt angle, the measured angle (thin black line at 10°) is represented as likelihood function (red), which is multiplied with the prior density (blue).** This yields the posterior density (yellow). The estimate is the mean of the posterior distribution (black dashed line). C and D are magnifications of A and B. A and C: For the local-likelihood model, prior, likelihood, and posterior are all symmetric. Thus, the posterior mean is attracted towards the prior mean (at zero). B and D: For the full-likelihood model, the likelihood function becomes asymmetric, leading to an asymmetric posterior distribution. Due to this asymmetry, the mean of the posterior is now larger than the measured angle, which results in a repulsive effect. Note that only the full-likelihood model is optimal for the chosen signal-dependent noise distributions.

compared the fits of the full-likelihood model to the data from Van Beuzekom & Van Gisbergen [7] shown in Fig 5 with four different prior densities: 1) von Mises density (fit see Fig 5), 2) wrapped Gaussian density, 3) mixture of Gaussian and uniform, and 4) $t$ location-scale density (see Methods). Since the first two models have 3 parameters each (two for noise, one for the prior), the other two 4 parameters (two for noise, two for the prior), model comparison was done using the Akaike Information Criterion (AIC), for which lower values indicate better models.

The model with the lowest AIC was the one with $t$ location-scale prior (AIC -384.5) followed by the wrapped normal prior (AIC -360.4), mixture Gaussian+uniform (-351.4) and the von-Mises prior (-351.1). The $t$ location-scale prior has a circular standard deviation of 15.0° with parameter $\nu$ = 94.0, which means that the shape of the density is close to a normal distribution. The model's utricular standard deviation is $\sigma_x$ = 0.051 g, the saccular SD $\sigma_z$ = 0.149 g. The fit is shown in Fig 10, and a comparison with Fig 5 shows that in contrast to using the von-Mises prior, here no clear E-effect is visible (Fig 10A), but the modeled response variability (Fig 10B red) more closely matches the data in the range of tilt up to about 135°. However, the difference between models in terms of root mean square error (RMSE; von Mises: 3.46°, $t$ location-scale: 2.63°) is not only due to the better fit of the variability but holds also when only SVV adjustments are considered (von Mises: 4.16°, $t$ location-scale: 3.39°).

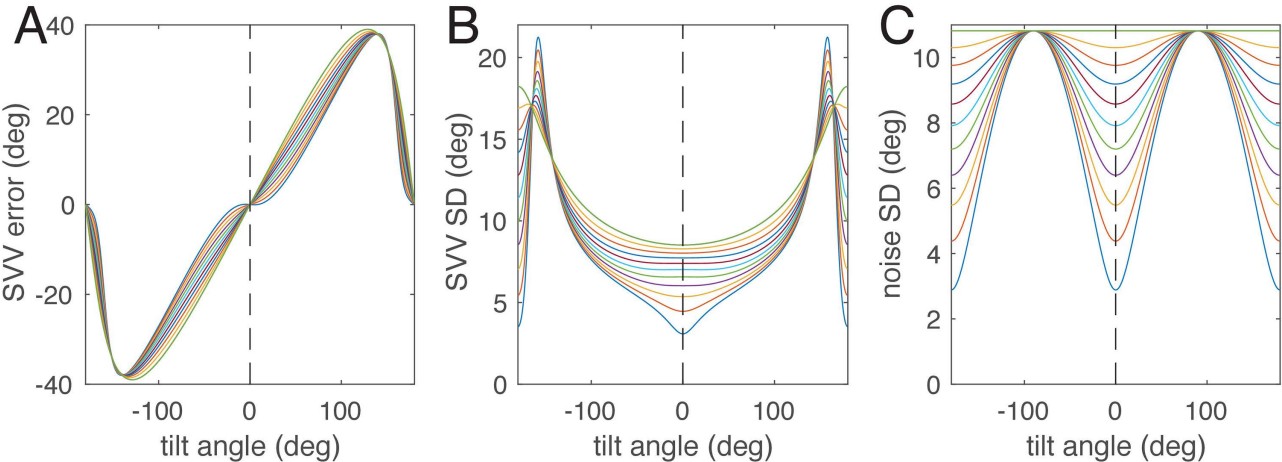

**Fig 8. Dependence on the relation between the saccular noise variance $\sigma_z^2$ and the utricular noise variance $\sigma_x^2$ for the SVV error (A), the standard deviation of the posterior distribution, and the standard deviation of the measurement noise (C).** The lowest thin blue line in B and C corresponds to the full-likelihood model shown in Figs 3, 4, and 6B. The thin green line on top corresponds to equal variances and results in measurement noise independent of tilt angle dependence (top line in C) and no E-effect in A.

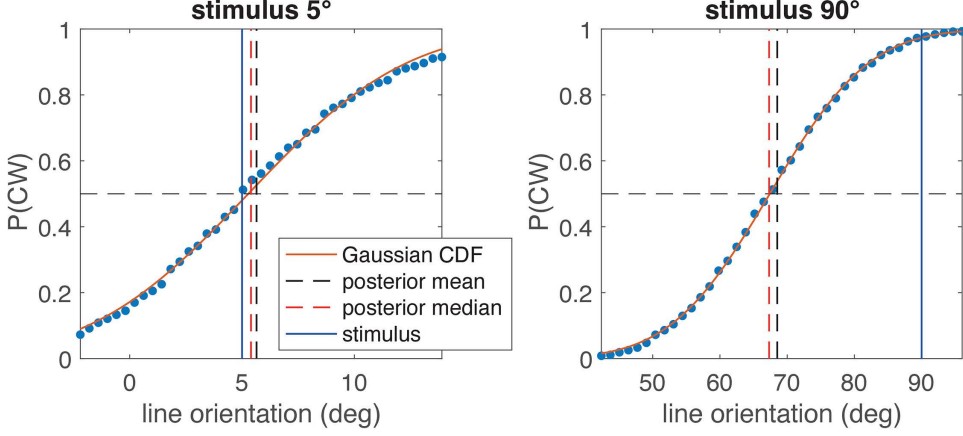

**Fig 9. Comparison of model simulations for adjustment of the SVV (vertical black dashed line, mean of the posterior) and the 2AFC method (vertical red dashed) for two head tilts (blue vertical line, left: 5°, right: 90°; note different scaling of the x-axis).** Blue dots are simulated proportions for the response "clockwise" plotted over indicator line orientation. The point of subjective equality, determined by a cumulative Gaussian fit, coincides with the theoretical circular half-mass median of the posterior (vertical dashed red line). For both stimulus orientations, the difference between mean (vertical dashed black line) and median (red line) is small, but for 5° it leads to a slightly decreased E-effect.

To further evaluate whether this result means that our perceptual system needs to exactly determine the correct shape of the orientation prior and the accurate otolith noise parameters to provide useful estimates of verticality, we generated 1e6 different tilt stimuli according to both models, i.e., using the fitted parameters for the generation of noisy stimuli from a distribution given by the respective prior. We then simulated the SVV adjustment with both models. When stimuli were generated according to the von Mises model in Fig 5 and estimated with the same (matching) model, the RMSE was 3.91°, when estimated with the t location scale model (Fig 10), which did not match the stimulus generation, the RMSE was 3.95°. This shows that even though the models have different outputs, the consequences of using the "wrong" prior

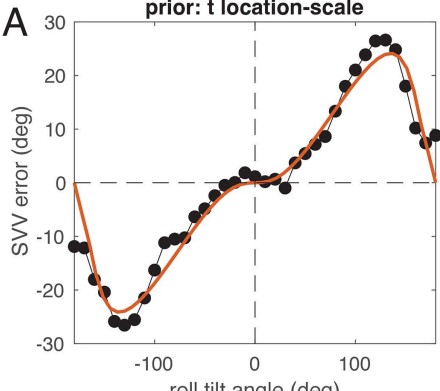
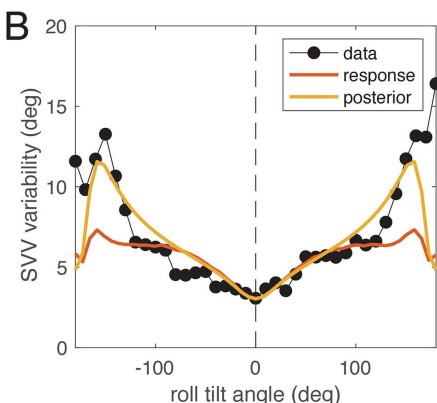

**Fig 10. The full-likelihood model (red) with _t_ location-scale prior and data (black dots) from Van Beuzekom & Van Gisbergen [7].** A: SVV error (black) plotted over roll tilt angle together with best fit of the model (red). B: experimental SVV variability (black) plotted over roll tilt angle together with the fitted response variability (red) and variability of the posterior distribution (yellow) from numerical model simulation, both expressed as angular standard deviation. Compare with Fig 5.

model are relatively small, which points towards a robustness of the estimation process even when the estimation model and the generative model do not match.

## Discussion

Here we show that the "anti-Bayesian" E-effect can be explained as Bayesian consequence of signal-dependent measurement noise resulting in asymmetric likelihood functions, which in turn causes the E-effect. Asymmetric likelihood functions can lead to repulsive bias [23]. Similar anti-Bayesian effects in orientation perception have previously been explained by asymmetric likelihood functions [1]. While Wei & Stocker [1] hypothesized efficient coding to be the reason for stimulus-dependent sensory noise, we showed here that the simple assumption of constant noise on the two otolith organs and its propagation through the internal computation of orientation with respect to gravity already causes tilt-dependent sensory noise, which in turn results in asymmetric likelihood functions causing the repulsive bias of the E-effect.

We furthermore show that our full Bayesian model, having only three free parameters, can fit not only average SVV adjustments covering the full 360° range from the literature (Fig 4A), but also closely mimics the variability found experimentally for these adjustments. The model's ability to match the experimentally found adjustments and their variability depended on the exact shape of the assumed prior distribution. However, we also showed that even if the perceptual process used an estimation procedure that did not match the generative process, e.g., a prior differing from the natural statistics of tilt angles, the resulting average perceptual error would not differ much. This points towards a considerable robustness of the perceptual system estimating verticality.

The result of the estimation stage of any Bayesian models is not a single value, but a posterior distribution, from which, in case of simulating a perceptual experiment like setting the SVV, a particular value must be chosen as estimate. Here we chose the mean of the posterior distribution, which minimizes the sum of the quadratic errors on a linear scale and the sum of the cosine distances on the circular scale. In the literature often the maximum of the posterior, the maximum-a-posteriori (MAP) estimate, is chosen. However, the MAP corresponds to a cost function that is 0 when the MAP exactly matches the stimulus and 1 otherwise. Such a 0–1 cost function is not well-suited for continuous estimation problems (see [26]), for example, because it means any error would be weighted equally, while it is more natural to put larger cost on larger errors. From Fig 6B and 6D it is obvious that taking the MAP as SVV angle would, in the illustrated case, not lead

to an E-effect, while taking the mean does. We also show that, in contrast to adjustments, when using a 2AFC procedure, the cost function is different and the resulting estimate is not the angular mean, but the circular half-mass median of the posterior distribution. This cost function leads to slightly different results for the SVV, which in our example (Fig 9) diminished the E-effect but enhanced the A-effect.

A recent comparison of SVV models [19], which focused on the effect of non-Gaussian priors, also tested a model that contained asymmetric likelihood functions but favored a previously proposed one with a Gaussian prior [17], which attributed the E-effect to an interaction of proprioception with the visual adjustment of the SVV, but assumed a symmetric likelihood as modelled in our suboptimal local-likelihood model. While it is completely reasonable to posit proprioceptive influences to affect the SVV (and might explain remaining differences between some of the SVV datasets in the literature and our model), our present work aims to emphasize that even without such additional inputs an optimal Bayesian SVV adjustment is completely compatible with both E- and A-effects. In addition, the E-effect does not depend on the exact shape of the prior, but on the asymmetric likelihood functions.

In the present model, the asymmetric likelihood functions are caused by stimulus dependent noise. To model stimulus dependent measurement noise, we chose a very simple assumption: additive Gaussian measurement noise on both otoliths, which is stimulus-independent, but differs for utricles and saccules, being larger in the latter. This choice is consistent with anatomical evidence: the utricle contains substantially more hair cells than the saccule [28]. Because the utricle is functionally most sensitive to tilts around 0°, while the saccule is most responsive near 90° [29], this asymmetry may explain the increase in angular noise at larger tilt angles [25]. Nonetheless, realistic neural noise might be better simulated by lognormal variability of otolith afferent firing rates. However, our simulations of lognormal afferent noise together with cross-striolar or commissural inhibition [30] resulted in signal-dependent angular variance that was almost indistinguishable from our initial assumption.

Udo de Haes [11] found stimulus independent variability of ocular counterroll, which would support our hypothesis of constant additive noise, since counterroll is determined predominantly by the utricles. Furthermore, neural variability of otolith afferents stays almost constant, at least for a range of ±0.5 g [31], corresponding to ±30° tilt. Concerning differences in measurement noise, a recent study [32] showed that perceptual thresholds of linear motion in the earth-horizontal plane along the utricular (y-) axis are much smaller (0.65 cm/s) than those along the saccular (z-) axis (1.2 cm/s). While this difference qualitatively supports our assumption, it would be interesting to see whether interindividual SVV characteristics correlate with differences in perceptual thresholds of linear motion.

An interesting consequence of the dependence of the E-effect on the complete likelihood function is that the stimulus dependence of the measurement distribution across the entire 360° range must be known for the estimation process. While it is conceivable for head tilt or other measures of orientation that the noise characteristics can be learned (see [33]) due to the closed circular range, it is practically impossible for magnitudes on open scales such as duration or distance. However, measurement of magnitudes on open scales does exhibit stimulus-dependent uncertainty. For duration perception this is known as scalar variability, and corresponds to Weber's law, which holds for most other magnitudes, including distances. Consequently, one possibility for magnitudes on open scales would be that the brain uses Weber's law as approximation of the true stimulus dependence of magnitudes, which in turn allows a good approximation of the respective likelihood functions. Such an approximation can be achieved by representing open-scale magnitudes on a logarithmic scale, as proposed previously (e.g. [34]).

In summary, our current model offers a Bayesian account of both the attractive Aubert and repulsive Müller biases in the subjective visual vertical, which is optimal, if sensory measurement noise depends on tilt angle. As possible explanation for tilt-dependent noise, we showed that assuming additive Gaussian noise for both otolith organs is sufficient, if noise levels differ for utricles and saccules. Thus, our model can account for the repulsive bias known as the E-effect without requiring additional contributions from ocular counterroll or proprioception originally suspected by Müller [6].

## Author contributions

**Conceptualization:** Stefan Glasauer.

**Data curation:** Stefan Glasauer, W. Pieter Medendorp.

**Formal analysis:** Stefan Glasauer.

**Funding acquisition:** Stefan Glasauer, W. Pieter Medendorp.

**Investigation:** Stefan Glasauer, W. Pieter Medendorp.

**Methodology:** Stefan Glasauer, W. Pieter Medendorp.

**Project administration:** Stefan Glasauer, W. Pieter Medendorp.

**Resources:** Stefan Glasauer, W. Pieter Medendorp.

**Software:** Stefan Glasauer.

**Supervision:** Stefan Glasauer, W. Pieter Medendorp.

**Validation:** Stefan Glasauer, W. Pieter Medendorp.

**Visualization:** Stefan Glasauer, W. Pieter Medendorp.

**Writing – original draft:** Stefan Glasauer.

**Writing – review & editing:** Stefan Glasauer, W. Pieter Medendorp.

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
