## [Decision Letter · Decision Letter 0]

16 Dec 2025

PCOMPBIOL-D-25-02021

Explaining attractive and repulsive biases in the subjective visual vertical

PLOS Computational Biology

Dear Dr. Glasauer,

Thank you for submitting your manuscript to PLOS Computational Biology. After careful consideration, we feel that it has merit but does not fully meet PLOS Computational Biology's publication criteria as it currently stands. Therefore, we invite you to submit a revised version of the manuscript that addresses the points raised during the review process. Please ensure all data and code is accessible when you resubmit.

We look forward to receiving your revised manuscript.

Kind regards,

Paul Bays

Academic Editor

PLOS Computational Biology

Hugues Berry

Section Editor

PLOS Computational Biology

**Journal Requirements:**

At this stage, the following Authors/Authors require contributions: Stefan Glasauer, and W. Pieter Medendorp. Please ensure that the full contributions of each author are acknowledged in the "Add/Edit/Remove Authors" section of our submission form.

Potential Copyright Issues:

i) Figures 1, and 4 appear to have been adapted from a previously published figure. Please provide written permission from the copyright holder to publish this under our CC-BY 4.0 license, or remove the figure / replace the image. Please note we do not recommend using standard request forms available on Publishers' websites, as they grant single use rather than republication under an open access license.

7) Kindly revise your competing statement in the online submission form to align with the journal's style guidelines: 'The authors declare that there are no competing interests.'

**Reviewers' comments:**

Reviewer's Responses to Questions

**Comments to the Authors:**

Reviewer #1: Glasauer & Medendorp propose a Bayesian model that accounts for both the attractive (Aubert) and repulsive (Müller) biases in subjective visual vertical (SVV) judgments across body tilt. The core contribution is the demonstration that tilt-dependent sensory noise in the otolith organs naturally produces asymmetric likelihoods, which in turn generate the repulsive bias at small tilts without invoking non-Bayesian mechanisms.

Overall, I think this is an interesting and clearly written manuscript. While asymmetric likelihoods have been previously proposed as explanations of ‘anti-Bayesian’ biases in other contexts, the fact that tilt-dependent estimation noise can be shown to directly arise from error propagation through the utricle–saccule coordinate transform here is neat. I have a couple of questions that I would like the authors to consider, along with some minor suggestions for improvement, listed below.

As the authors point out, repulsive biases only arise in the model if the posterior mean is used. While this estimator is appropriate for modelling performance on a continuous adjustment SVV, does it raise the possibility that different task structures (e.g. binary choice, is line tilted towards/away from body relative to vertical) might yield different results? Being able to account both for situations in which repulsive biases do and do not occur would add weight to the proposed explanation.

How robust are modelling outcomes to the introduction of non-Gaussian otolith noise? Are asymmetric likelihoods and repulsive biases retained under more realistic neural noise statistics?

Minor points:

P1. ‘in the upright position, the [mean] SVV error is close to zero’

Figure 2. At printed scale, the repulsive E-effect is not clearly visible as stated in the legend.

Reviewer #2: The authors present a Bayesian model of attractive and repulsive bias during a specific perceptual inference task. Although other models of the behavioural pattern already exist, theirs is the first normative (Bayesian) model to jointly account for both the attractive and repulsive biases with minimal additional assumptions.

I found the manuscript convincing, employing a principled approach to a well-defined problem. I have no technical concerns regarding the provided models. I do think the manuscript could benefit from more explicit contextualization and presentation of relevant information (described in detail below), but this is overall minor.

In particular, I think it would suit the authors to clarify the following:

- Although the task itself is well-presented, not much motivation is provided as to why the task is particularly notable in the broad setting of perceptual inference tasks. (Is it just that it yields both attractive and repulsive biases? Is there anything else?)

- Interpreting the model fits requires identifying the signatures of the A- and E-effects on plots like the ones in Figure 2. I found that doing this took some effort. I suggest that, given how important visual interpretation of these effects is, the authors find a visual way of demonstrating or illustrating them (e.g., via highlighting or by an additional panel with schematized plots)

- It might be beneficial to have a two-sentence primer on the biology early in the paper (it is possible to pick this up incidentally, but it would have made for a smoother reading experience to have it earlier)

- What’s the point of fitting the Bayesian models to the simulated outputs from the idiotropic model when you later fit to participant data (which is presumably what actually matters)?

- It may be worth explicitly distinguishing between “measured value” (meaning, the output of the vestibular sensors) and “observed value” (meaning, what the participants actually reported) — since these are both, in a sense, “measured”, some might be confused by this. This is especially relevant in Figure 7 where “measured”, as far as I can tell, actually is meant in the latter sense whereas generally throughout the article it is mostly meant in the former sense.

- In figure 4 it may be worth annotating what each entry of each matrix encodes. For example, each entry of the prior is P(theta), each entry of the likelihood matrix is P(measurement | tilt angle), and each entry of the posterior matrix is P(SVV estimate | measurement)

- Since the response variability estimated by your model (i.e., based on the posterior dispersion) is systematically higher than observed response variability, it may be worth a discussion of elements that may account for this potentially-significant difference

**Have the authors made all data and (if applicable) computational code underlying the findings in their manuscript fully available?**

The PLOS Data policy requires authors to make all data and code underlying the findings described in their manuscript fully available without restriction, with rare exception (please refer to the Data Availability Statement in the manuscript PDF file). The data and code should be provided as part of the manuscript or its supporting information, or deposited to a public repository. For example, in addition to summary statistics, the data points behind means, medians and variance measures should be available. If there are restrictions on publicly sharing data or code —e.g. participant privacy or use of data from a third party—those must be specified.requires authors to make all data and code underlying the findings described in their manuscript fully available without restriction, with rare exception (please refer to the Data Availability Statement in the manuscript PDF file). The data and code should be provided as part of the manuscript or its supporting information, or deposited to a public repository. For example, in addition to summary statistics, the data points behind means, medians and variance measures should be available. If there are restrictions on publicly sharing data or code —e.g. participant privacy or use of data from a third party—those must be specified.requires authors to make all data and code underlying the findings described in their manuscript fully available without restriction, with rare exception (please refer to the Data Availability Statement in the manuscript PDF file). The data and code should be provided as part of the manuscript or its supporting information, or deposited to a public repository. For example, in addition to summary statistics, the data points behind means, medians and variance measures should be available. If there are restrictions on publicly sharing data or code —e.g. participant privacy or use of data from a third party—those must be specified.requires authors to make all data and code underlying the findings described in their manuscript fully available without restriction, with rare exception (please refer to the Data Availability Statement in the manuscript PDF file). The data and code should be provided as part of the manuscript or its supporting information, or deposited to a public repository. For example, in addition to summary statistics, the data points behind means, medians and variance measures should be available. If there are restrictions on publicly sharing data or code —e.g. participant privacy or use of data from a third party—those must be specified.

Reviewer #1: **No:**Model code not currently availableModel code not currently availableModel code not currently availableModel code not currently available

Reviewer #2: Yes

PLOS authors have the option to publish the peer review history of their article (what does this mean?). If published, this will include your full peer review and any attached files.). If published, this will include your full peer review and any attached files.). If published, this will include your full peer review and any attached files.). If published, this will include your full peer review and any attached files.

...

Reviewer #1: No

Reviewer #2: No

**Figure resubmission:**
---

## [Editor Report · Decision Letter 1]

23 Mar 2026

Dear Prof. Dr. Glasauer,

We are pleased to inform you that your manuscript 'Explaining attractive and repulsive biases in the subjective visual vertical' has been provisionally accepted for publication in PLOS Computational Biology.

Best regards,

Paul Bays

Academic Editor

PLOS Computational Biology

Hugues Berry

Section Editor

PLOS Computational Biology

---

## [Editor Report · Acceptance letter]

PCOMPBIOL-D-25-02021R1

Explaining attractive and repulsive biases in the subjective visual vertical

Dear Dr Glasauer,

I am pleased to inform you that your manuscript has been formally accepted for publication in PLOS Computational Biology. Your manuscript is now with our production department and you will be notified of the publication date in due course.

With kind regards,

Anita Estes
